# Sitting Posture during Prolonged Computer Typing with and without a Wearable Biofeedback Sensor

**DOI:** 10.3390/ijerph18105430

**Published:** 2021-05-19

**Authors:** Yi-Liang Kuo, Kuo-Yuan Huang, Chieh-Yu Kao, Yi-Ju Tsai

**Affiliations:** 1Department of Physical Therapy, College of Medicine, National Cheng Kung University, Tainan 701, Taiwan; yiliangkuo@mail.ncku.edu.tw; 2Department of Orthopedics, National Cheng Kung University Hospital, Tainan 701, Taiwan; hkyuan@mail.ncku.edu.tw; 3Department of Rehabilitation, Sengkang Community Hospital 1 Anchorvale Street, Singapore 544835, Singapore; jeiyukao@gmail.com; 4Institute of Allied Health Sciences, College of Medicine, National Cheng Kung University, Tainan 701, Taiwan

**Keywords:** sitting posture, computer users, wearable sensor, spine, biofeedback

## Abstract

Prolonged sitting combined with an awkward posture might contribute to the increased risks of developing spinal pain. Maintaining an upright sitting posture is thus often suggested, especially nowadays when people spend longer periods in the sitting posture for occupational or leisure activities. Many types of assistive devices are commercially available to help computer users maintain an upright sitting posture. As the technology advances, wearable sensors that use microelectromechanical technology are designed to provide real-time biofeedback and promote adjusting posture actively. However, whether such wearable biofeedback sensors could assist adjusting sitting posture in computer users during prolonged typing remains unknown. This study aimed to investigate the effects of a wearable biofeedback sensor on maintaining an upright sitting posture. Twenty-one healthy young adults were recruited and performed a 1-h computer typing task twice, with and without using the active biofeedback device. The sagittal spinal posture during computer typing was measured using a three-dimensional motion analysis system. Using the wearable biofeedback sensor significantly decreased the neck flexion (*p* < 0.001), thoracic kyphotic (*p* = 0.033), and pelvic plane (*p* = 0.021) angles compared with not using the sensor. Computer users and sedentary workers may benefit from using wearable biofeedback sensors to actively maintain an upright sitting posture during prolonged deskwork.

## 1. Introduction

In modern society, people spend a significant amount of time sitting for occupational or leisure activities [1,2]. Prolonged sitting combined with an awkward posture, such as a forward-leaning head and increased thoracic kyphosis, increases the demands of the spinal muscles and joints, which might contribute to the increased risks of developing spinal pain in sedentary workers [3,4]. Previous studies showed that 17.7–63.0% and 23–34% of office workers experienced neck pain and back pain during the last 12 months, respectively [4,5,6]. Extended computer use during daily learning activities and recreation also contributes to the high prevalence rates of neck pain and/or back pain among college students [7,8]. Chronic spinal pain can lead to both physical and psychological problems for individuals, including a reduced range of motions, muscle weakness, disability, depression, and reduced quality of life. Back pain and neck pain have been recognized as the leading global cause of disability in most countries [9]. It results in an enormous economic burden for both individuals and societies due to healthcare costs, decreased productivity, work absenteeism, lost wages, and work compensation [10,11]. The total cost of back pain and neck pain in 2016 was estimated to be 1345 billion USD, which was the highest healthcare expenditure among 154 health conditions in the United States [11].

Studies have demonstrated that a slump posture or forward head posture resulted in increased external loading and muscle activities on the spine, and these postural changes were also associated with spinal pain [12,13,14,15]. Therefore, to maintain an upright sitting posture is commonly suggested [16]. An upright postural alignment involves “a minimal amount of stress and strain and which is conductive to maximal efficiency in the use of the body” [17]; however, maintaining an upright sitting posture is not easy [18,19]. Falla et al. reported that participants without neck pain or disorders progressively slouched (increased thoracic kyphosis) during a computer task as short as 10 min [18]. Claus et al. found that healthy participants required visual and verbal feedback to reproduce an upright sitting posture [19]. Therefore, many types and designs of chairs and assistive devices have been developed to promote an upright sitting posture [20,21,22,23].

Annetts et al. [20] compared the effects of different types of chairs on spinal angles of the cervical and lumbopelvic regions. Grondin et al. [22] investigated the use of a lumbar support pillow in maintaining a thoracolumbar posture. These chairs and assistive devices are considered passive devices because their designs are aimed to passively support the user to maintain an upright sitting posture. However, previous studies demonstrated that different types of chairs had various effects on the postural alignment and did not consistently produce an upright posture across spinal regions. The effects of the passive devices on maintaining an upright posture were inconsistent. Ergonomically designed chairs also have the disadvantages of being large in size, not easy to carry, and have limited usage in certain locations. Those chairs and passive devices may not be suitable for individuals with different anthropometric characteristics. 

In contrast to passive devices, active devices are developed to provide feedback and promote postural adjustment actively. Celenay et al. showed an improvement in thoracic posture after an 8-week training program wearing a harness with an electronic sensor placed between the scapulae and waist [24]. An auditory alarm was given once the individual slouched and the sound could only be stopped while returning to the previous posture. Furthermore, Yoo et al., investigating the effects of auditory feedback from a chair during a short period of computer work (15–20 min), observed immediate effects on the muscle activities and kinematics of the head and neck [25,26]. The effects of these active devices depend on whether the users actively adjust their posture when feedback is received. Although previous studies have reported acceptable outcomes with these active devices, their prototype designs are preliminary and bulky, with many wires, making them impractical for everyday use [24,26]. 

As the technology advances, wearable sensors that use microelectromechanical technology are designed to provide real-time biofeedback and promote an upright posture actively. Such active biofeedback sensors are wearable devices that can be worn discretely on clothing and thus have the advantages of a compacted size, easy-use, portable use, and without a location limitation. The wearable sensor can be set up to vibrate whenever the user slouches. As a result, the user can actively correct his or her posture. 

The effectiveness of different types of ergonomic training, including the use of assistive devices, was evaluated in a recent overview of available systematic reviews [27]. While 21.6% of the systematic reviews included supported the implication of ergonomic training to reduce physical demand and musculoskeletal symptoms among workers, 78.4% of systematic reviews found disproof or insufficient evidence to suggest a benefit or harm of the intervention for clinical practice. The authors attribute inconsistent evidence to factors such as erroneous identification of ergonomic risk factors and lack of scientific knowledge about the appropriate combination of ergonomic training. Nevertheless, wearable sensors were not included in any of the systematic reviews. Whether wearable biofeedback sensors could promote an upright sitting posture remains unclear. 

Computer users and sedentary workers often seek solutions to maintain an upright sitting posture. Therefore, this study aimed to investigate the effects of a wearable biofeedback sensor in actively modifying the spinal posture during a prolonged computer typing task. The results of this study would provide useful information for health professionals who face inquiry for the usefulness of wearable biofeedback sensors and for people who experience musculoskeletal discomfort during prolonged computer work.

## 2. Materials and Methods 

### 2.1. Study Design

This was a one-group quasi-experimental study. A single group of participants was measured twice, with and without using a wearable biofeedback device, while performing a computer typing task. The independent variables were the condition and time. The dependent variables were spinal angles. Data collection was conducted at the laboratory in the university.

### 2.2. Participants

Healthy young adults were recruited from the university campus. The inclusion criteria were as follows: (1) age between 20–25 years; and (2) willing and able to give informed consent for participation in the study. All participants were free from spinal pain and had no previous injury or surgery in the spine and abdominal regions. This study was approved by the investigators’ institutional review board. Written informed consent was obtained from all participants. 

For the a priori power analysis, we used the G*power 3.1 software [28] to determine the sample size for this study. The minimum sample size was estimated based on the reported difference of sitting posture after 8-week spinal postural training using a biofeedback posture-correction device [24]. The required sample size for a statistical power of 0.8 and a two-tailed α of 0.05 for two dependent means was 6 and 16 for thoracic and lumbar angles, respectively. Considering that this study aimed to investigate the effects of a wearable biofeedback sensor during a single session of a computer typing task, the effect size of this study may be smaller than the calculated values from the literature. Therefore, we enrolled a convenient sample of 21 participants in this study. 

### 2.3. Instrumentation

Lumo Lift (Lumo Bodytech Inc., Palo Alto, CA, USA) was used in this study, which includes an accelerometer-based sensor (approximately 4.4 cm long × 2.6 cm wide × 1.3 cm high), a magnetic clasp (approximately 1.5 cm long × 1.5 cm wide × 0.3 cm high), and an universal serial bus charging dock (Figure 1). This wearable biofeedback sensor can be worn with the sensor portion underneath the clothes, directly below the clavicle, and midway between the sternal notch and the acromion process. It provides vibratory biofeedback by comparing the acceleration due to gravity with the user’s downward acceleration. Based on our pilot data, this wearable biofeedback sensor measures about 5° of angular change from the preset posture in the sagittal plane. 

A three-dimensional motion analysis system with six infrared cameras (Vicon Motion Systems Ltd., Oxford, UK) was used to measure the spinal posture at a sampling rate of 100 Hz during the computer typing tasks. Reflective markers were attached onto the bilateral canthus and tragus; suprasternal notch and spinal processes of T1, T3, T9, T11, L1, L2, L4, and L5; bilateral anterior superior iliac spines; and posterior superior iliac spines. The Vicon motion analysis system is a reliable instrument for postural measurement [29].

### 2.4. Procedure

The participants completed a baseline survey, which included sex, side of the dominant hand, age, height, mass, and duration of daily sitting time and computer use. After completing the baseline survey, the investigator attached reflective markers to specific anatomical landmarks of participants while standing. The investigator also directly attached the sensor to the skin, below the clavicle, and midway between the sternal notch and the acromion process. Then, all the participants performed a 1-h computer typing task twice, with and without the sensor in a random order.

For the computer typing tasks, the participants sat on a backless, armless, and height-adjustable wooden chair with their hips and knees flexed to 90°. The computer workstation consisted of a standard computer desk (75-cm height), a standard keyboard, a mouse, and a 20-inch monitor. The monitor was placed in front of the participants, with the upper edge of the screen adjusted to the eye level. The participants were instructed to copy and type from an electronic book. After completing the first typing task, the participants were allowed to rest as long as they wished until they were ready for the second typing task.

Before commencing the computer typing task, the participants were verbally and manually guided by the investigator to sit with their scapulae slightly retracted and the thoracolumbar spine extended. This upright sitting posture was used to preset the wearable biofeedback sensor as the target posture. The participants were then instructed to actively adjust and resume the upright sitting posture whenever they received vibratory biofeedback from the sensor during the computer typing task.

### 2.5. Data Analysis

The kinematic data were digitized using Vicon Nexus 1.8.5 (Vicon Motion Systems Ltd.) and then filtered by a low-pass fourth-order Butterworth filter of 4 Hz using MATLAB software (The MathWorks, Inc., Natick, MA, USA). Spinal posture was evaluated on the basis of the joint angles or segment inclination angles, including head tilt, neck flexion, upper cervical, lower cervical, thoracic, lumbar, and pelvic plane angles (Figure 2) [30,31].
(a)Head tilt angle: the segment of the mid-point of canthus and the mid-point of tragus relative to the horizontal plane.(b)Neck flexion angle: the segment formed by the mid-point of tragus and spinous process of T1 relative to the frontal plane.(c)Upper cervical angle: the angle between the mid-point of canthus, the mid-point of tragus and the T1 spinous process.(d)Lower cervical angle: the angle between the mid-point of tragus, the T1 spinous process and the suprasternal notch.(e)Thoracic angle: the angle between the segments of T1–T3 and T9–T11.(f)Lumbar angle: the angle between the segments of L1–L2 and L4–L5.(g)Pelvic plane angle: the angle between the segment of the mid-point of anterior superior iliac spines and the mid-point of posterior superior iliac spines relative to the horizontal plane.

Decreasing head tilt, upper cervical, and lower cervical angles indicate flexion, and decreasing neck flexion, thoracic, and lumbar angles indicate extension. A negative pelvic plane angle indicates an anterior tilt of the pelvis. The neck flexion angle is commonly used to quantify the forward head posture [32,33], with a greater angle suggesting a more forward head posture. A greater thoracic angle indicates increased thoracic kyphosis. The last 30-s angle data of the 5th, 15th, 25th, 35th, 45th, and 55th minute of 1-h typing were averaged for statistical analysis.

Descriptive statistics using means and standard deviations for continuous data and frequency for nominal data were used to describe the study sample. For each spinal angle, separate two-way analysis of variance with repeated measures was used to examine the statistically significant differences between the two conditions (with and without the sensor) at six time points. Significant main effects were followed up using Bonferroni’s correction post-hoc analyses. Effect sizes (r) based on the F-values were calculated [34], and classified as small (r = 0.2), medium (r = 0.3), and large changes (r ≥ 0.5) [35]. All data were analyzed using SPSS version 20 (IBM corp., Armonk, NY, USA) with the level of significance set as *p* < 0.05.

## 3. Results

Twenty-one participants (12 women, 9 men; age 23.33 ± 2.9 years; weight 61.4 ± 10.0 kg; height 167.0 ± 9.0 cm) enrolled in this study. Participants spent an average of 9.7 ± 3.2 h/day sitting and 7.0 ± 2.9 h/day using computer. Spinal angles during the computer typing tasks with and without using the wearable biofeedback sensor across different time points are shown in Table 1. Significant condition effects were observed for the neck flexion, upper cervical, lower cervical, thoracic, and pelvic plane angles (*p* < 0.05, Table 1). On average, the neck flexion, upper cervical, lower cervical, thoracic, and pelvic plane angles were significantly smaller with the wearable biofeedback sensor than without. The mean differences in the neck flexion, upper cervical, lower cervical, thoracic, and pelvic plane angles between the two conditions were 2.8° (95% confidence interval (CI): 1.4°–4.2°, r = 0.68), 2.8° (95% CI: 1.0°–4.6°, r = 0.59), 1.2° (95% CI: 0.2°–2.0°, r = 0.53), 1.9° (95% CI: 0.2°–3.6°, r = 0.46), and 2.2° (95% CI: 0.4°–4.1°, r = 0.49), respectively. No significant difference was observed in the other spinal angles (*p* > 0.05, Table 1). In addition, no significant time or interaction effects were identified (all *p* > 0.05, Table 1).

## 4. Discussion

Maintaining an upright sitting posture is often suggested for preventing excessive external mechanical loading and muscle activations on the spine, especially nowadays, when people spend longer periods in the sitting posture for occupational or leisure activities. Young adults use computer extensively for learning activities and recreation, which greatly increases their risk of developing spinal pain [7,8]. In a prospective study with 8-year follow-up, experiencing low back pain in youth was found to correlate with low back pain in adulthood [36]; therefore, it is important to identify effective approaches for maintain an upright sitting posture in the younger population. This study investigated the immediate effect of a wearable biofeedback sensor on the spinal posture of young adults during a prolonged computer typing task. 

Our findings showed that using the active biofeedback sensor resulted in significantly smaller neck flexion, upper cervical, lower cervical, thoracic, and pelvic plane angles. These angular changes suggest that participants had a less forward head, less thoracic kyphosis, and a less posterior tilted pelvis. In other words, the participants were able to maintain a more upright sitting posture during the 1-h computer typing task when the wearable biofeedback sensor was used. A forward-leaning head is the most commonly recognized postural fault that is related to neck pain in adults [15]. A concern is whether the small postural changes resulting from the use of the wearable biofeedback sensor (range 1.2°–2.8°) would have any clinically meaningful benefits although the calculated effect sizes are medium to large (range 0.46–0.68). Conversely, these small postural changes cumulated throughout a workday may amplify the effect of the active biofeedback sensors. Further investigation is required to determine the long-term effects of active biofeedback sensors.

Posture is controlled through the integration of sensory information and motor output. The sense of the position and movement of our body parts that is provided by the somatosensory system signals us when and how to respond to the environment [37]. However, proprioceptive sensation can be influenced by musculoskeletal disorders. Previous studies showed that people with spinal pain demonstrated impaired proprioception when assessed with postural repositioning or motion perception threshold tests [38,39]. The ability to maintain an upright sitting posture during prolonged computer work may be difficult if there is no sufficient and accurate intrinsic feedback through proprioceptive sensations. Ribeiro et al. [40] addressed the importance of extrinsic feedback to help execute or improve motor performance in situations where motor improvements are difficult to achieve. Extrinsic feedback is thought to enhance the somatosensory system and restore optimal motor control [41,42]. Our study provides preliminary evidence to support the use of the wearable biofeedback sensor to provide extrinsic feedback and facilitate active postural adjustment. Active postural adjustment might also enhance the conscious awareness of the body posture, which is the subjective phenomenological aspect of proprioception. Higher postural awareness is associated with reduced clinical symptoms in patients with chronic pain [43]. The wearable biofeedback sensor has the advantage of being miniaturized, more portable, and more suitable for everyday use over other active postural devices used in studies [25,26]. In addition, active postural adjustment through extrinsic feedback might train the muscles involved in the task and decrease unnecessary muscle activation [44], which subsequently may lower the risk of musculoskeletal symptoms related to a poor posture.

Spinal posture involves the alignment of multiple joints. Kuo et al. found a significant chain of correlations between sagittal spinal angles in the sitting posture [31]. Decreased forward leaning of the neck was associated with the downward tilt of the head. The results of this study did not support previous findings. The use of the wearable biofeedback sensor did not alter the head tilt despite significant changes in other spinal regions within the kinematic chain. The biomechanical link between the neck flexion and head tilt angles was possibly interrupted by the computer typing task. The participants were required to tilt their head downward and upward to gaze at the keyboard and computer screen while typing. Therefore, the participants might have tilted their head upward while maintaining a less forward lean of the neck posture with the use of the wearable biofeedback sensor.

This study has several limitations. First, only the immediate effect was investigated. Participants in a previous study reported that vibrotactile feedback was easily notable but disturbing compared with graphical and physical types of feedback [45]. Some participants in this study commented that they might turn off the vibrotactile feedback in a long-term study. The compliance factor would definitely influence the effectiveness of the wearable biofeedback sensor for everyday use. The long-term effects of wearable biofeedback sensors must be further determined with a well-designed randomized controlled study. Second, measurement of spinal angles using the motion analysis system required the participants to remove the T-shirt or tank top for skin marker attachment. The wearable biofeedback sensor used in this study was directly attached to participants’ skin below the clavicle with adhesive tapes instead of being attached under the clothes using a magnetic clasp. The wearable sensor may not accurately detect a slouched posture if it is worn under loose-fitting clothes. The effect of the wearable biofeedback sensor for everyday use may be compromised. In addition, in this study, we analyzed the effects of only one type of active biofeedback device. Therefore, the results of this study may not be inferred to all active devices available in the market.

## 5. Conclusions

The wearable sensor using biofeedback is able to assist maintaining an upright sitting posture during a single session of prolonged computer typing. Using the wearable biofeedback sensor significantly decreased the neck flexion, thoracic kyphotic, and pelvic plane angles in healthy young adults compared with not using the sensor. Computer users and sedentary workers may benefit from the use of wearable biofeedback sensors to actively maintain an upright sitting posture during prolonged computer work; however, the long-term effects of wearable sensors on prevention and treatment of spinal pain requires further investigation.

## Figures and Tables

**Figure 1 ijerph-18-05430-f001:**
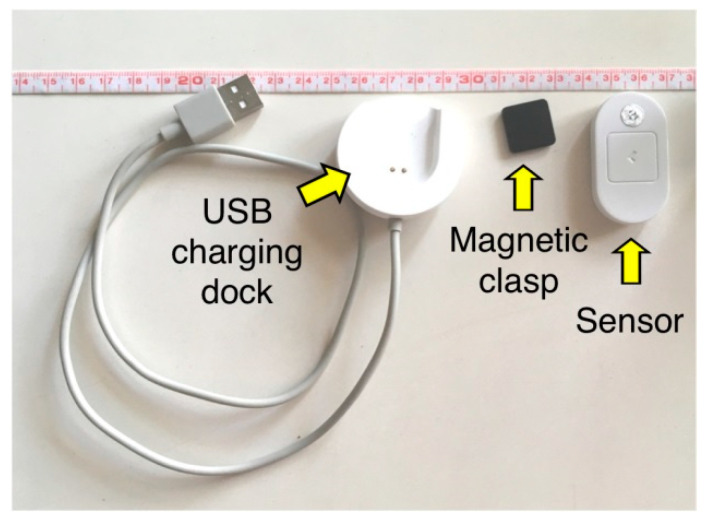
The wearable biofeedback sensor used in this study.

**Figure 2 ijerph-18-05430-f002:**
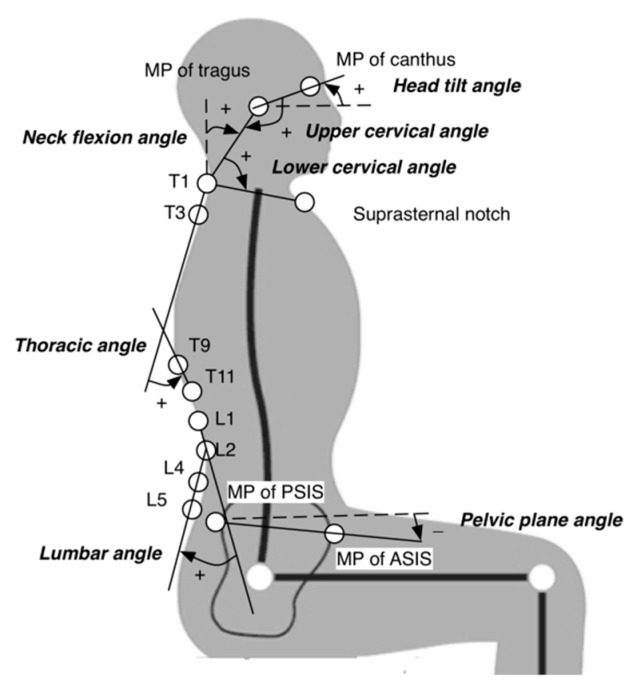
Placement of reflective markers and angle definitions. Decreasing head tilt, upper cervical, and lower cervical angles indicate flexion. Decreasing neck flexion, thoracic, and lumbar angles indicate extension. A negative pelvic plane angle indicates an anterior tilt of the pelvis. MP: midpoint; ASIS: anterior superior iliac spine; PSIS: posterior superior iliac supine.

**Table 1 ijerph-18-05430-t001:** Descriptive and inferential statistics of the spinal angles under two conditions across six time points (N = 21).

Angles	Device	T_5_	T_15_	T_25_	T_35_	T_45_	T_55_	Condition (C)	Time (T)	Interaction (C × T)
Head tilt	Without	12.7 ± 7.4	11.7 ± 8.0	11.6 ± 8.8	11.4 ± 8.6	13.7 ± 9.5	11.3 ± 9.4	*p* = 0.434	*p* = 0.582	*p* = 0.079
	With	12.2 ± 8.9	12.5 ± 9.7	13.1 ± 9.7	12.9 ± 8.7	12.3 ± 9.4	12.9 ± 10.2			
Neck flexion	Without	59.3 ± 5.3	60.2 ± 6.2	59.7 ± 6.5	59.8 ± 6.3	59.0 ± 6.6	60.3 ± 6.2	*p* < 0.001 *	*p* = 0.619	*p* = 0.201
	With	57.2 ± 5.7	57.4 ± 5.7	56.8 ± 5.8	56.5 ± 5.8	57.1 ± 6.0	56.6 ± 5.9			
Upper cervical	Without	155.1 ± 8.2	154.9 ± 9.0	154.1 ± 9.7	153.9 ± 8.9	155.7 ± 10.5	154.4 ± 10.7	*p* = 0.004 *	*p* = 0.650	*p* = 0.334
	With	151.6 ± 7.2	152.3 ± 8.5	152.3 ± 8.1	151.7 ± 7.7	151.6 ± 8.1	151.6 ± 8.8			
Lower cervical	Without	62.3 ± 8.6	61.8 ± 8.0	61.5 ± 7.9	61.4 ± 8.3	62.2 ± 9.5	61.4 ± 8.6	*p* = 0.012 *	*p* = 0.688	*p* = 0.149
	With	60.3 ± 7.6	60.7 ± 7.5	60.9 ± 8.1	60.4 ± 7.7	60.5 ± 7.5	60.9 ± 8.1			
Thoracic	Without	28.9 ± 7.8	29.7 ± 8.0	28.8 ± 9.0	29.0 ± 8.3	29.3 ± 9.6	29.6 ± 8.0	*p* = 0.033 *	*p* = 0.816	*p* = 0.613
	With	27.2 ± 8.1	27.6 ± 8.4	27.2 ± 8.2	27.5 ± 9.5	27.9 ± 9.1	26.6 ± 8.7			
Lumbar	Without	8.6 ± 4.9	9.4 ± 4.9	9.3 ± 4.6	8.9 ± 5.2	8.9 ± 4.5	8.4 ± 5.2	*p* = 0.217	*p* = 0.094	*p* = 0.516
	With	7.2 ± 4.8	9.0 ± 5.0	8.1 ± 5.4	7.1 ± 4.7	8.1 ± 5.0	8.5 ± 5.1			
Pelvic plane	Without	7.3 ± 6.6	7.0 ± 7.3	7.7 ± 6.8	6.8 ± 6.4	8.8 ± 8.8	6.7 ± 8.4	*p* = 0.021 *	*p* = 0.566	*p* = 0.501
	With	5.7 ± 6.3	5.2 ± 7.1	5.8 ± 6.5	4.1 ± 5.6	4.8 ± 6.8	5.5 ± 5.5			

Values are expressed as the mean ± standard deviation. * The post-hoc analysis indicates any statistically significant difference between the two conditions (without and without the sensor).

## Data Availability

The data presented in this study are available on request from the corresponding author.

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
