# Peer review of "Sitting Posture during Prolonged Computer Typing with and without a Wearable Biofeedback Sensor"

_ijerph, 2021, doi:10.3390/ijerph18105430_

Round 1

Reviewer 1 Report

The authors have answered my comments clearly. Congratulations.

Author Response

Thank you for your constructive suggestions and kind words.

Reviewer 2 Report

Authors should clarify how they can assess posture using a single accelerometer

Table 1
The statistics should also highlight the post hoc analysis

Reviewer 3 Report

This was a well-designed and well-presented study. I have only a few minor comments/suggestions.

Line 68: "chairs or lumbar supports also have the disadvantages of relatively huge in size" - Should be "disadvantages of being relatively huge in size" - Also, relative to what? Should state this. Finally, chairs are obviously much bigger than other lumbar supports. May be best to say that these products are much larger than small, wearable devices (such as the one in this study).

Line 94 - "contribute" - I think what you mean here is "attribute"

Line 285-286 - "Some participants commented that they might turn off vibrotactile feedback in a long-term study." - Participants in your study or in the study cited in the previous sentence? Maybe make this clearer 
